# Soft Actor-Critic with Inhibitory Networks for Faster Retraining

## Abstract

Reusing previously trained models is critical in deep reinforcement learning to speed up training of new agents. However, it is unclear how to acquire new skills when objectives and constraints are in conflict with previously learned skills. Moreover, when retraining, there is an intrinsic conflict between exploiting what has already been learned and exploring new skills. In soft actor-critic (SAC) methods, a temperature parameter can be dynamically adjusted to weight the action entropy and balance the explore × exploit trade-off. However, controlling a single coefficient can be challenging within the context of retraining, even more so when goals are contradictory. In this work, inspired by neuroscience research, we propose a novel approach using inhibitory networks to allow separate and adaptive state value evaluations, as well as distinct automatic entropy tuning. Ultimately, our approach allows for controlling inhibition to handle conflict between exploiting less risky, acquired behaviors and exploring novel ones to overcome more challenging tasks. We validate our method through experiments in OpenAI Gym environments.

## 1 Introduction

In reinforcement learning (RL), for all but the simplest tasks, agents behavior must be optimized with respect to multiple goals and constraints (Sutton & Barto, 2018). It is common in practice for new objectives and constraints to arise after already having trained an agent on existing ones. In order for the behavior of the agent to account for these additional constraints, retraining is required. Within the context of deep RL, retraining an agent to new constraints gives problems in balancing exploitation of previously learned skills with learning new skills. In this paper, we present a method for efficiently retraining an agent to acquire new skills in a similar environment.

Existing approaches to solving the problem of learning new skills include the use of hierarchical RL structures (Dayan & Hinton, 1993; Barto & Mahadevan, 2003), such as the options framework (Sutton et al., 1999; Comanici & Precup, 2010), universal value functions (Schaul et al., 2015), option-critic (Bacon et al., 2017), FeUdal networks (Vezhnevets et al., 2017), and data-efficient hierarchical RL (Nachum et al., 2018). Other methods use multiple policies and value functions each of which are optimized for simple objectives that can then be composed to achieve complex objectives (Van Seijen et al., 2017; Sahni et al., 2017; Haarnoja et al., 2017; Hansen et al., 2020; Barreto et al., 2020). In this paper we propose to address the problem through the use of multiple value functions to provide a complex evaluative input to a single policy network. By applying different value functions in a state dependent fashion, the reward provided to the policy network during training can remain the same as in prior training when appropriate, and can switch to a different reward when the situation indicates new constraints or goals.

The mechanism we propose to govern the output of the composite value function is based upon neuroscientific research on inhibitory control (Diamond, 2013). The brain uses inhibition to interrupt ongoing goal-directed action when unexpected events or conflicts arise. The horse-race model of Logan et al. (2015), which describes the behavioral response to a goal and its inhibition in terms of dual processes, is supported by many behavioral and neurobiological studies (Verbruggen & Logan, 2009; Shenoy et al., 2011; Ide et al., 2013; Schall et al., 2017). We implement this inhibitory concept in soft actor-critic (SAC) algorithm (Haarnoja et al., 2018a) by using an additional value network (inhibitory), as opposed to retraining the previously learned value network (ongoing), to

learn the new skill evaluation. Additionally, we propose the use of an inhibitory policy network to control inhibition. We call this SAC with inhibitory networks approach SAC-I. SAC-I is distinct from previous value composition work (Haarnoja et al., 2017; Van Niekerk et al., 2019), since it proposes a specific mechanism to train and compose value networks and generate a single policy network focused on fast and improved retraining. We hypothesize that retraining of RL agents can be accelerated by creating independent and mutually inhibitory evaluative processes that will change the reward function used during learning in a state dependent manner. SAC provides two important features for the SAC-I: a replay buffer that can be partitioned into episodic memories related to each evaluative-learning process (Botvinick et al., 2019) and an automated entropy estimation (Haarnoja et al., 2018b), which allows computing two separate temperature parameters and exploring actions differently.

There are two main contributions in this work. **First**, we develop the SAC-I architecture for accelerated retraining, that encompasses the use of inhibitory networks for the control of multiple evaluative networks. This approach modifies SAC methods by separating the learning process, which includes training multiple value functions, storing episodic replay buffers, estimating distinct temperature parameters, and learning an inhibition policy when necessary (Section 3). **Second**, we provide a detailed validation showing the different components of SAC-I and its improvements over SAC in two modified environments from OpenAI Gym (Section 4). The LunarLanderContinuous-v2 with a bomb appearing randomly resembles the classic stop-signal paradigm (Logan et al., 2015; Verbruggen & Logan, 2009) in inhibitory control studies. A mixed version of the BipedalWalkerHardcore-v3, highlights the out-performance of SAC-I over SAC, which is not able to successfully solve the task.

## 2 BACKGROUND AND RELATED WORK

### 2.1 MAXIMUM ENTROPY REINFORCEMENT LEARNING

In RL, knowing the best way to explore while exploiting is non-trivial, environment-dependent, and still an active area of research (Hong et al., 2018). Maximum entropy RL theory provides a principled way to address this particular challenge, and has been a key element in many of the recent RL advancements, providing improved exploration and faster learning (Thomas, 2014; Schulman et al., 2017a; Haarnoja et al., 2017; 2018a;b; Ziebart, 2010). Given a Markov decision process (MDP) with a set of states $S$, a set of actions $A$, a transition function $T$ and a reward function $R$, forming a tuple $< S, A, T, R >$ (Puterman, 1994), a stochastic policy $\pi : S \to A$ is a mapping from states to probabilities of selecting each possible action, where $\pi(a|s)$ represents the probability of choosing action $a$ given state $s$. In maximum entropy RL, as in an MDP, the goal is to find the optimal policy $\pi^*$ that provides the highest expected sum of rewards, while additionally maximizing the entropy of each visited state, leading to the expression (Ziebart, 2010):

$$\pi^* = \arg\max_\pi \sum_t \mathbb{E}_{(s_t,a_t)\sim\rho_\pi}[r_t + \alpha\mathcal{H}(\pi(\,\cdot\,|s_t))], \tag{1}$$

where $\alpha$ is the temperature parameter that controls the stochasticity of the optimal policy, $\rho_\pi$ is the state-action marginal of the trajectory distribution induced by the policy, $r_t$ is a shorthand for the environmental reward $r(s_t, a_t)$ at time $t$, and $\mathcal{H}(\pi)$ represents the entropy of the policy, $\mathbb{E}[-log(\pi(a|s))]$. This approach allows a state-wise balance between exploitation and exploration. For states with high reward, a low entropy policy is permitted while, for states with low reward, high entropy policies are preferred, leading to greater exploration. The discount factor $\gamma$ is omitted in the equation for simplicity since it leads to a more complex expression for the maximum entropy case (Thomas, 2014). But it is required for the convergence of infinite-horizon problems, and it is included in our final algorithm.

### 2.2 SOFT ACTOR-CRITIC

Soft actor-critic (SAC) (Haarnoja et al., 2018a) is one of the most successful maximum entropy RL methods and has become a common baseline algorithm in most of the RL libraries, outperforming state-of-the-art methods Haarnoja et al. (2018a;b). Like the deep deterministic policy gradient (DDPG) approach (Lillicrap et al., 2016), SAC is a model-free and off-policy method, using a replay buffer, where the policy and value functions are approximated using neural networks. In addition,

it incorporates a policy entropy term into the objective function facilitating exploration, similar to soft Q-learning (Haarnoja et al., 2017). Similar to trust region policy optimization (TRPO) (Schulman et al., 2015) and proximal policy optimization (PPO) Schulman et al. (2017b), SAC uses a stochastic policy and is known to be more stable than DDPG. In short, SAC combines the best of DDPG (sample efficiency) and TRPO/PPO (stability through stochastic policies). As expressed in Equation 1, the SAC policy/actor is trained with the objective of maximizing the expected cumulative reward and the action entropy at a particular state. The critic is the soft Q-function and, following the Bellman equation, is expressed by: $Q(s_t, a_t) = r_t + \gamma \mathbb{E}_{s_{t+1} \sim p}[V(s_{t+1})]$, where $p$ represents the state transition probability, and the soft value function is parameterized by the Q-function: $V(s_{t+1}) = \mathbb{E}_{a_t \sim \pi}[Q(s_{t+1}, a_{t+1}) - \alpha \log \pi(a_{t+1}|s_{t+1})]$. The soft Q-function is trained to minimize the following objective function given by the mean squared error between predicted and observed state-action values:

$$J_Q = \mathbb{E}_{(s_t, a_t) \sim \mathcal{D}} \left[ \frac{1}{2} \big( Q(s_t, a_t) - (r_t + \gamma \mathbb{E}_{s_{t+1} \sim p}[\bar{V}(s_{t+1})]) \big)^2 \right], \tag{2}$$

where $\mathcal{D}$ denotes the replay buffer, and $\bar{V}$ is the target value function (Mnih et al., 2015). Finally, the policy is updated to minimize the KL-divergence between the policy and the exponentiated state-action value function (Haarnoja et al., 2018b), and can be expressed by:

$$J_\pi = \mathbb{E}_{s_t \sim \mathcal{D}} \left[ \mathbb{E}_{a_t \sim \pi}[\alpha \log \pi(a_t|s_t) - Q(s_t, a_t)] \right]. \tag{3}$$

## 2.3 INHIBITORY CONTROL

Inhibitory control, also known as response inhibition, is a critical component of the executive functions and refers to the ability to modify ongoing actions in response to unexpected and dynamically changing task demands (Aron, 2007; Diamond, 2013). In Shenoy et al. (2011), inhibitory control is formalized as a rational decision-making problem, and a computational model using Bayesian inference and stochastic control tools is proposed and validated by behavioral data from humans and animals. Using a widely adopted paradigm known as the *stop-signal task* (Logan et al., 2015), authors show that the optimal policy, whether to go or stop, systematically depends on accumulating sensory evidence, which supports the hypothesis that the brain is implementing statistically optimal decision-making (Shenoy et al., 2011). Figure 1 depicts the two processes involved in the stop-signal task. The Go process starts with a go signal followed by a stop signal which triggers the stop process. The stop process will dominate when its activation is larger than the Go process activation. The key assumption is that both processes are stochastically independent, as supported by behavioral studies (Verbruggen & Logan, 2009). Stop-signal reaction time (SSRT) is defined as the time necessary to respond to the stop stimulus. In further work using functional MRI, the anterior cingulate cortex, a region in the brain implicated in a variety of cognitive control functions, is shown to activate proportionally to a Bayesian prediction error between predicted and observed events resembling the temporal-difference methods in reinforcement learning (Ide et al., 2013). These previous computational models indicate that when dealing with unexpected events such as the stop signal the brain implements a dual-process model driven by prediction error and responds to the conflict in an optimal way.

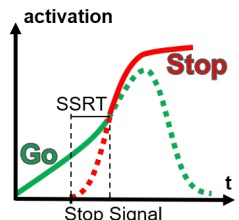

Figure 1: Dual process model in the stop-signal task (SST).

## 2.4 RELATED WORK

**Value function composition**. In value function composition (Haarnoja et al., 2017; Van Niekerk et al., 2019), the goal is to model a new task by composing value functions previously trained on sub-tasks. In Todorov (2009), they show that for linearly solvable MDPs (Todorov, 2007), pretrained value function estimators can be optimally composed and solved. The composition is a union of tasks, an "OR" composition, and is defined by taking the softmax over the component reward signals. Van Niekerk et al. (2019) extends this result to the standard and entropy regularized RL settings. Haarnoja et al. (2017) defines a composition rule that approximately solves the intersection of tasks in the entropy regularized setting, an "AND" composition, with the composed signal as an average over the constituent rewards. Despite the similarities, our approach is fundamentally different since it composes a previously learned value function with one which is newly trained, and moreover does not involve combining their value estimates.

**Hybrid reward architecture**. In Hybrid Reward Architecture (HRA) (Van Seijen et al., 2017), the goal is to learn a complex task by decomposing its reward, and training separate value functions for each component. The total reward is replaced by an equivalent representation as the sum, an "AND" composition, of decomposed constituent rewards. They show that HRA learns more efficiently than the deep Q-network algorithm (DQN) (Mnih et al., 2015) when both algorithms have otherwise identical network architectures. SAC-I is similar to HRA in the training of multiple Q networks using different rewards, however essentially different since it does not aim to combine the reward values. In our approach, Q networks are trained independently and used to provide specialized action values depending on the state, which is defined by the inhibition rule.

**Transfer learning in RL**. Broadly speaking, transfer learning in RL consists of transferring the knowledge gained in one task to improve the learning performance in a related, but different task (Taylor & Stone, 2009; Lazaric, 2012). This knowledge can be some type of learned representation (Rusu et al., 2016b), reward shaping (Brys et al., 2015), demonstration (Schaal, 1996), model dynamics (Ammar et al., 2012), or policy (Rusu et al., 2016a). Our work is within the large field of transfer learning, however we are primarily focused on transferring learned value and policy functions among identical aspects of a task, while learning new skills (value functions) and retraining the previously learned policy within the similar environment.

## 3 METHODS

### 3.1 SAC-I: SAC WITH INHIBITORY NETWORKS

Inhibitory control is traditionally defined as the ability to stop ongoing or planned cognitive or motor processes, overriding impulsive or habitual responses (Aron, 2007). One possible RL implementation is at the executive level by switching between two policies using a hierarchical architecture. However, this approach requires having to train multiple policies, and a more complex hierarchical model. We propose a computationally less expensive approach and

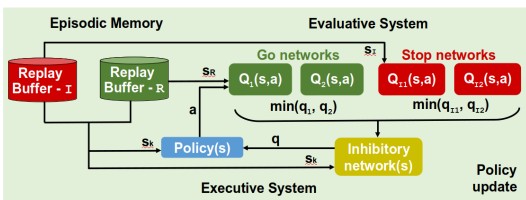

Figure 2: SAC-I policy update.

do the switching at the Q network level, since it will ultimately drive the change of the habitual response or action. We implement inhibitory control in this broader sense at a higher cognitive level, where the action execution (policy) is updated by the evaluative processes that precede execution.

We apply inhibitory control to the SAC algorithm by having multiple, competing value functions (critics) that take turns depending on the current task demand. Rather than having a Q network that learns the Go task and later the Stop task, we keep the previous Q network that knows to evaluate the Go task and train a new Q-I network that learns the Stop task. Thus, both Q networks are trained independently. Here, the term "inhibitory" refers to a new evaluative process replacing an ongoing one. In Figure 2, we depict the SAC policy update with inhibitory networks. Go networks estimate ongoing evaluation (which led to pre-trained skills), while the Stop networks estimate the value of the new event, leading the policy to learn new skills. Additionally, we introduce the use of an *inhibitory policy network* to learn the inhibition policy responsible for deciding how and when to use the outcomes from the multiple Q networks. Alternatively, to avoid having to train an additional policy, a parametric state-based *inhibition rule* can be implemented using knowledge about the environment or updated constraints. We use Q networks to estimate all the value functions, following the implementation in Haarnoja et al. (2018b), and the clipped double-Q learning trick, introduced in twin delayed DDPG (TD3) (Fujimoto et al., 2018) to avoid their overestimation.

### 3.2 EPISODIC MEMORY THROUGH SELECTIVE REPLAY BUFFER

In neuroscience research, episodic memory refers to the brain's ability to recollect past experiences, and is an important component of learning (Tulving, 2002). In recent years, it has been applied to DRL as a non-parametric framework to retrieve past successful experiences to improve sample efficiency (Blundell et al., 2016; Lin et al., 2018; Botvinick et al., 2019) or to avoid catastrophic forgetting (Isele & Cosgun, 2018). In this work, we implement episodic memory in its simplest form; a partition of the state space according to an inhibition rule or policy that yields a partition

of the replay buffer. Let $S = \{S_R, S_I\}$ be a partition of the state space. The corresponding replay buffer partition is $\mathcal{D} = \{\mathcal{D}_R, \mathcal{D}_I\}$ where set membership of a tuple $(s_k, a, r_k, s')$ is parameterized by $s_k$ for $k \in \{R, I\}$ where $s_R \in S_R$ and $s_I \in S_I$. For ease of notation, $s' = s_{t+1}$, and the next state can belong to any of the partitions, $s' \in S$. Note that inhibitory states have an associated inhibitory reward $r_I$ and regular states have the associated reward $r_R$. These corresponding rewards are stored in the replay buffers but the replay buffers are not parameterized by them. The loss function of each Q network is computed using memories from the corresponding replay buffer as expressed by:

$$J_{Q_k} = \mathbb{E}_{(s,a) \sim \mathcal{D}_k} \left[ \frac{1}{2} \left( Q_k(s, a) - (r_k + \gamma \, \mathbb{E}_{s' \sim p}[\bar{V}(s')]) \right)^2 \right] \qquad \text{for } k \in \{R, I\}. \tag{4}$$

Notice that if we use a single replay buffer that contains both the regular and the inhibitory rewards $(s, a, r_R, r_I, s')$, and sample a tuple containing the regular $(s, a, r_R, 0, s')$ or the inhibitory $(s, a, 0, r_I, s')$ rewards depending on the updated Q network, we would have sparser rewards particularly for the Q-I network. Therefore, having separate memories promotes faster learning (as shown in Figure 4) and, importantly, also allows the estimation of separate entropy parameters.

### 3.3 Automated Dual Entropy Estimation

The SAC algorithm is sensitive to the $\alpha$ temperature changes depending on the environment, reward scale and training stage, as shown in the initial paper (Haarnoja et al., 2018a). To address this issue, the same authors propose to automatically adjust the temperature parameter by formulating the problem with a dual objective; maximize entropy while satisfying a minimum entropy constraint (Haarnoja et al., 2018b). The goal is still maximizing the cumulative expected reward (Equation 1), but the average entropy of the policy is now constrained by a minimum value. The full derivation of the dual optimization problem is given in Haarnoja et al. (2018b). In practice, the optimization is performed recursively as follows: at time $t$, given the current estimate $\alpha_t$, the optimal policy $\pi_t^*$ is estimated as described in Equation 1. Subsequently, given $\pi_t$, the $\alpha_t$ is approximated using a neural network. While in the standard SAC, there is a single $\alpha$ parameter, in SAC-I, we propose estimating two separate $\alpha$ temperature parameters, $\alpha_R$ and $\alpha_I$, to allow distinct action entropy for the previously learned and the new skills, respectively. This is implemented by training two separate $\alpha$ networks, with regular states $s_R \in \mathcal{S}_\mathcal{R}$ distinguished from inhibitory states $s_I \in \mathcal{S}_I$, with losses given by:

$$J_{\alpha_k} = \mathbb{E}_{a \sim \pi}[-\alpha_k \log \pi(a \mid s_k) - \alpha_k \mathcal{H}_0], \qquad \text{for } k \in \{R, I\}, \tag{5}$$

where $\mathcal{H}_0$ is the minimum expected entropy. The policy loss function is composed of two terms:

$$J'_\pi = \sum_{k \in \{R,I\}} \mathbb{E}_{s_k \sim S_k} \left[ \mathbb{E}_{a \sim \pi}[\alpha_k \log \pi(a|s_k) - Q_k(s_k, a)] \right]. \tag{6}$$

### 3.4 Inhibitory Policy Network

In our proposed SAC-I algorithm (Figure 2), for the cases in which a state-dependent inhibition rule is not defined, we propose training an inhibitory policy network $\pi_I$. This network can be trained as an automated *hard switch* or a *soft modulator* between the regular and inhibitory Q networks. The inhibitory policy network is a stand-alone agent with its own Q networks (not shown in Figure 2), however it shares the same replay buffers. The loss functions are the standard ones as defined in Equations 2 and 3. Ultimately, the goal of the inhibitory policy network is to maximize the environment's reward by learning to choose between the Go and Stop networks and/or by modulating the Stop network. We show the implementation of both cases next in Section 4.

SAC-I is summarized in Algorithm 1.

## 4 Experiments and Results

In order to show two different use-cases of SAC-I algorithm, as well as to evaluate it as a way to speed up training during transfer learning, we use continuous tasks from the Box2D simulator, OpenAI Gym (Brockman et al., 2016), LunarLanderContinuous-v2 and BipedalWalkerHardcore-v3. Importantly, we include custom modifications to them to emulate the scenarios in which retraining is necessary. By adding a random bomb in the LunarLander task, we show the advantages of using

---

**Algorithm 1:** Soft Actor Critic with Inhibitory Networks (SAC-I)

---

Initialize $Q_{R_1}, Q_{R_2}, Q_{I_1}, Q_{I_2}$, policy $\pi$, policy $\pi_I$, $\alpha_R$ and $\alpha_I$ networks parameters;
Initialize the target $\bar{Q}_{R_1}, \bar{Q}_{R_2}, \bar{Q}_{I_1}$ and $\bar{Q}_{I_2}$ networks weights;
Initialize the replay buffers $\mathcal{D}_R$ and $\mathcal{D}_I$;
**for** *each episode* **do**
    **for** *each environment step* **do**
        Given $s_t$, sample $a_t$ from $\pi(s_t)$ and $(s_{t+1}, r_t)$ from the environment;
        Use an *inhibition rule* or inhibitory policy $\pi_I$ to classify $s_t$;
        if $s_t \in S_R$, push $(s_t, a_t, r_{R_t}, s_{t+1})$ to $\mathcal{D}_R$;
        else if $s_t \in S_I$ push $(s_t, a_t, r_{I_t}, s_{t+1})$ to $\mathcal{D}_I$;
    **end**
    **for** *each gradient step* **do**
        Sample a batch of memories from from $\{\mathcal{D}_R, \mathcal{D}_I\}$;
        **for** $k \in \{R, I\}$ **do**
            Update $Q_{k_1}$ and $Q_{k_2}$ (Equation 4), $\alpha_k$ (Equation 5), and $\bar{Q}_{k_1}$ and $\bar{Q}_{k_2}$ (soft-update);
        **end**
        Update the policy network $\pi$ (Equation 6);
        If an inhibitory policy network is used, update $\pi_I$ and the associated $Q$ networks;
    **end**
**end**

---

SAC-I, compared to standard SAC, when retraining with conflicting goals, similar to what happens in a stop-signal task (Logan et al., 2015), i.e. stopping an ongoing action (to land) whenever an unexpected event occurs (to avoid bomb). In the experiments with BipedalWalkerHardcore-v3, we train agents in a simpler version of the task (BipedalWalker-v3), and retrain them in the more complex task. We show how SAC-I can help transfer learning and adjust inhibitory control. In all the experiments provided in this section, we used five random seeds to account for the variability during training and compare agents trained across a fixed number of steps. All the hyperparameters are available in Table 1 (Appendix).

## 4.1 LUNARLANDERCONTINUOUS WITH BOMB

**Environment**. The original version of LunarLanderContinuous-v2 is modified in order to include a bomb that appears randomly within a region above the landing pad (Figure 3). Like the original version, it includes the environment reward (moving from the top of the frame to the landing pad: 100-140 points, each leg contact: +10, crashing: -100, successful landing: +100, firing engine: -0.3 per frame), but additionally it includes a penalty for hitting the bomb (-150) and a time penalty (-0.1 per frame) to motivate landing as quickly as possible (like in SST). Importantly, the bomb coordinates are included in the observation state only after the bomb appears so the agent does not know about its existence beforehand. Further details can be found in Appendix.

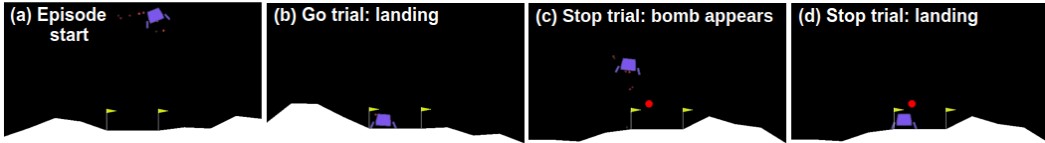

Figure 3: LunarLanderContinuous-v2 with Bomb. (a) Episode starts with the lander placed randomly on top of the frame around the center. (b) Go trial: refers to an episode without a bomb, and it is exactly as the original environment. (c) Stop trial: refers to episodes in which a bomb randomly appears close to the landing pad (between the two flags). (d) It shows a successful landing during a stop trial after avoiding hitting the bomb. By default, the bomb appears in 50% of the episodes.

**Experimental design**. Initially, a standard SAC agent is trained in the original environment LunarLanderContinuous-v2. This agent, which we will call "baseline agent", takes about 250K steps to reach an average cumulative reward of 200 and is able to successfully land in most of the episodes. All the network weights are transferred to the retrained agent. The Q-I network is trained from scratch. In this experiment, the inhibition rule/policy works as a switch between the

regular and inhibitory Q networks. We show both cases, user-defined inhibition rule and learned inhibitory policy network. Experiments are performed in a single machine with Intel Core i7-9850H CPU@2.60Hz x 12, Quadro RTX3000, RAM 16GB. The average rewards reported in the figures represent the average of the episode reward, including the bomb penalty, over the last 100 episodes. Agents are trained for a fixed number of 2K episodes, approximately 500K steps.

**Advantage of retraining**. In this experiment, we show how the performance of a standard SAC agent is boosted by retraining it using weights from the baseline agent, which already knows how to safely land. In order to learn bomb avoidance skills, we shaped the original reward with an additional penalty given by the expression $r_{bomb.proxy} = -1e4 \times (d_b - 0.3)^4$, where $d_b$ is defined as the agent's distance to the center of the bomb $(x_b, y_b)$. The idea is to have a field-type avoidance mechanism and give a penalty proportional to bomb proximity. Average reward results are depicted in Figure 4, orange and blue lines. For retraining, all the network weights are loaded from the baseline agent and updated in the new task with bomb. The retrained agent (blue line) clearly learns to complete the task (land and avoid the bomb) faster than the agent trained from scratch (orange line). In about 300K steps, it starts to avoid the bomb and at 500K steps is mostly able to complete the task successfully. The agent from scratch takes $> 1.25M$ steps to learn the task.

**Effect of using episodic memory and dual alpha in SAC-I**. We parse out the distinct contributions of different components of SAC-I by training agents with and without episodic memory and the dual $\alpha$ temperature parameters. In these versions, we use a user defined inhibition rule given by $y > y_b$ and $d_b < 0.3$. If these conditions are met, the inhibitory Q-I network is used with the same shaping used for the SAC agent, $r_I = r_{bomb.proxy}$. Results are shown in Figure 4. The SAC-I vanilla agent (green line) is trained without episodic memory and with a single $\alpha$ parameter. The SAC-I with episodic memory (purple line) is trained with separate experience replay buffer for the inhibitory states, but uses only a single entropy parameter $\alpha$. The SAC-I agent with episodic memory and dual $\alpha$ (red line) has faster learning, reaching an average reward of 200 after 300K steps, and highlighting the importance of these components[1].

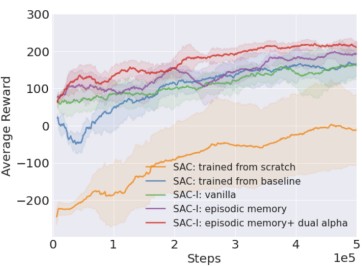

Figure 4: Comparison of different SAC and SAC-I agents.

**SAC-I performance for different conflict levels**. The brain's inhibitory control is known to be modulated by levels of conflict between ongoing processes (Braver et al., 2001). For instance in the SST, changing the frequency of stop signals, which is associated with expectation, alters the response time to go signals. In a similar way, we investigate whether and how the frequency of stop trials (episodes with bomb) impacts the agent's learning, other than the overall performance which is expected to decrease for increased occurrence of bombs. We hypothesize that the bomb frequency will impact SAC-I less than SAC because it trains separate Q networks for different skills involved in the task (i.e. landing and bomb avoidance). Both SAC and SAC-I agents are trained with varying bomb frequencies, and results are shown in Figure 5. As expected, the overall performance

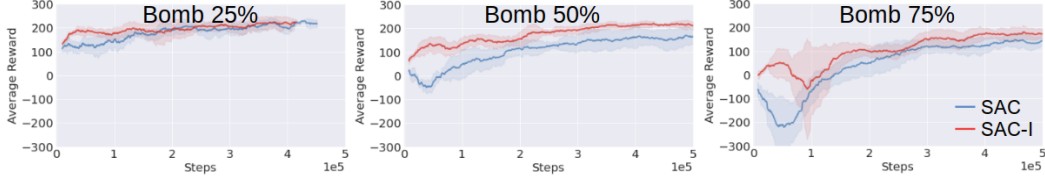

Figure 5: Performance of SAC-I and SAC agents for different bomb frequencies.

decreases for higher frequency. SAC-I agent reaches an average reward of 200 in about 200K, 300K and 600K for frequencies 25%, 50% and 75%, respectively. For the SAC agent, it is clear that bomb frequency not only affects performance but also the training progression. For example, a significant drop in performance is observed around step 50K for Bomb 75% as well as for 50%, although less. This likely happens because learning to avoid the bomb interferes with its initial ability to land.

---

[1]Further SAC-I agents are all trained with episodic memory and dual $\alpha$ parameters estimation.

Interestingly, although both agent's performances converge asymptotically, the difference is larger for Bomb 50% case, in which the uncertainty is the highest. This is likely because SAC and SAC-I agents learn different strategies to avoid the bomb. While the SAC agent adopts an optimal global strategy accordingly to different bomb frequencies (i.e, if bomb is frequent, slow down when task starts), the SAC-I agent keeps the same landing strategy independent of bomb frequency and learns to avoid the bomb when it appears.

**Training without reward shaping**. Further, we examined the training of SAC and SAC-I agents without using any reward shaping for the bomb avoidance. Both agents use the same reward $r = r_0 + r_{bomb}$, where $r_0$ is the original reward and $r_{bomb}$ is a sparse penalty for hitting the bomb. Results[2] are shown in Figure 6. Surprisingly, the SAC* agent without shaping (yellow line) performed better than the one with, meaning that shaping is negatively impacting its training. In contrast, the SAC-I* agent without shaping (gray line) successfully learns the task with similar speed to the one with shaping (red line), likely because it keeps separate critic networks to evaluate landing and bomb avoidance. In further experiments,

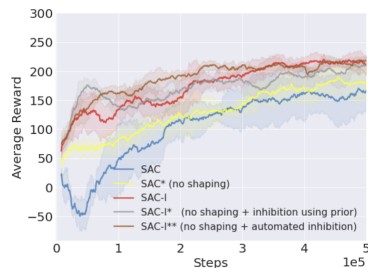

Figure 6: SAC and SAC-I agents with and without shaping.

we observed that different reward shaping can boost SAC-I's performance even more, but not SAC (see results with conservative shaping in Appendix, Figure 10).

**Learning when to inhibit**. In this experiment, we show an agent trained with an *inhibitory policy network* SAC-I** and compare it to an agent trained with an inhibition rule SAC-I*. The SAC-I* agent (gray line) uses a simple inhibition rule ("whenever bomb appears") to switch between the regular and inhibitory Q networks (Figure 2). While, the SAC-I** agent (brown line) learns an inhibition policy, i.e. the best timing to inhibit. This SAC-I** agent (brown line) performs as good as the SAC-I* and outperforms the SAC agents (Figure 6).

## 4.2 BIPEDALWALKERHARDCORE-V3

In this evaluation, we show a different use of inhibitory networks, to control the magnitude of inhibition. To create the retraining scenario, first, we train a standard SAC agent in an easier environment BipedalWalker-v3 (Figure 7a), in which the goal is to walk through a plain terrain, and use its weights to retrain SAC and SAC-I agents in BipedalWalkerHardcore-v3 (Figure 7b-d). In the BipedalWalker task, agents naturally get a negative "inhibitory" reward whenever they are stuck. We use SAC-I to learn as well as to weight that negative reward.

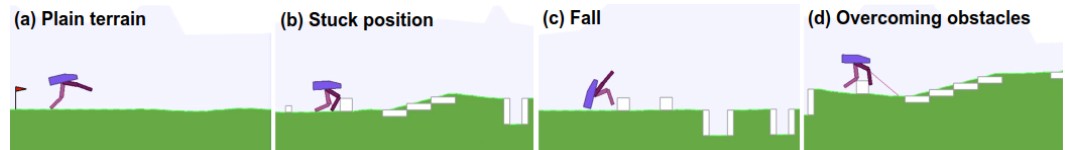

Figure 7: BipedalWalker environment. (a) Plain terrain: agent has to learn to walk forward; (b) Stuck position: agent starts to receive negative reward since it is spending energy without moving forward; (c) Fall: agent can fall because it loses balance, stumbles itself, or fall into a hole; (d) Overcoming obstacles: the agent learns avoiding the holes or going over the blocks.

The BipedalWalkerHardcore-v3 is a challenging task, known to be unsolvable for many of the simpler non-recurrent DRL architectures or model-free RL methods (Wei & Ying, 2021). We solve the task using a standard two-layer dense-network architecture, and adopt two strategies: removing the fall-penalty and creating a cumulative version of the task reward. Otherwise, the task is unsolvable with SAC algorithm (see Figure 11 in Appendix).

**Learning how to inhibit**. In Figure 8, we show the training performance of SAC and two versions of SAC-I agents learning BipedalWalkerHardcore-v3. All agents are retrained from baseline. The SAC agent reward consists of the raw environment reward $r_0$, but with a cumulative reward and without the fall penalty[3], expressed by $r = r_0 + r_{stuck} - r_{fall}$.

---

[2]SAC and SAC-I agents from Figure 4 are also shown to facilitate comparison, using the same color coding.

[3]Mimicking life, harsh fall penalty seems to block learning. See Figure 11 in Appendix.

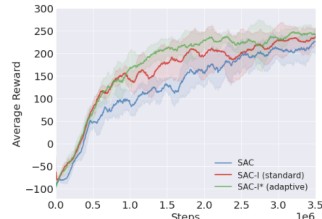

The cumulative reward is defined by $r_{stuck} = \sum_{i=0}^{5} r_{t-i}$, accounted if it is negative, i.e. the agent is at a "stuck" state. The cumulative reward provides a less noisy feedback. The primary goal of SAC-I (standard) is to separately train a Q-I critic network specialized on modeling the new reward structure, i.e. $r_I = r_0 + r_{stuck} - r_{fall}$. The SAC-I* agent (adaptive) learns an inhibitory policy network to estimate the weight $w$ so that $r_{I*} = r_0 + w \times r_{stuck} - r_{fall}$. This SAC-I* agent with adaptive inhibition (green line) outperforms the other agents, and the difference is highlighted in the mixed version of the task (see further Figure 9).

Figure 8: Performance of SAC and SAC-I agents.

**Mixed version of BipedalWalkerHardcore-v3**. To replicate the scenario in which there are go and stop episodes, we retrain the baseline agent on a modified task mixing both the BipedalWalker-v3 (Go trial) and BipedalWalkerHardcore-v3 (Stop trial). For each episode, we randomly choose in which environment the agent should perform. Interestingly, this makes the learning even more challenging, because the obstacles are sparser in time and the agent has to explore and exploit while learning the easy and hard versions of the task simultaneously. In Figure 9, we show that using inhibitory networks is critical to successfully learn the mixed task. Both agents are retrained using the same reward structure as presented in the previous section. Although the task gets easier as the percent of go trials is increased, from 10% to 30%, we observe that it is more difficult to learn the mixed version of the task since there are less stop trials to learn from (for instance, compare the two plots in the middle column). Also, we observe some training instability for 70% stop trial, likely because the interference between learning the Stop and Go trials (3rd column). Interestingly, we observe that the SAC-I* agent with adaptive inhibition has more stability across the stop training sessions. For Stop 90%, the averages of the reward standard deviation are 47.5 and 27.5 for SAC-I and SAC-I* agents, respectively. For Stop 70%, the averages are 56.3 and 31.7, respectively. Finally, we observe that, unlike the SAC agent, only the SAC-I agents are able to successfully learn the stop trials (2nd column). Note, they are trained using the same reward structure (previous section).

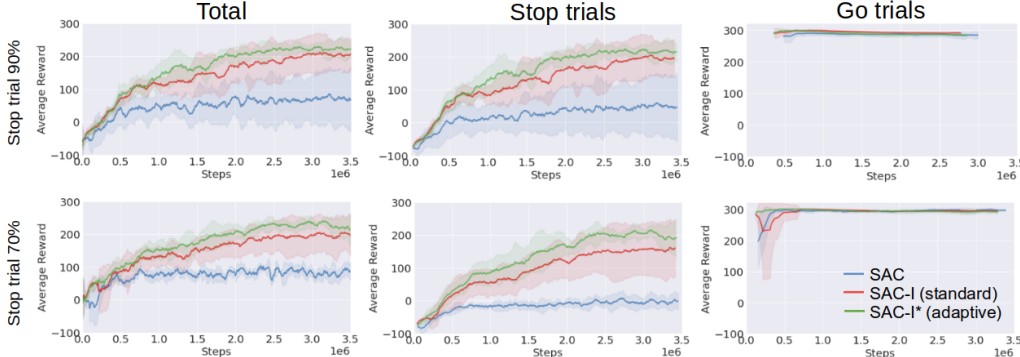

Figure 9: SAC and SAC-I agents performance in the mixed version of the BipedalWalkerHardcore-v3. First row shows results for the task with 90% Stop trials (hardcore), while the second, with 70%. From the left to the right, rewards are averaged across all, only Stop and only Go trials.

## 5 CONCLUSIONS

In this work, we draw a parallel between neuroscience research in inhibitory control and reinforcement learning of competing value functions. We propose an SAC algorithm using inhibitory networks with a particular focus on retraining the agent to acquire a new skill, while still exploiting previously learned abilities. Through our experiments, we show that SAC-I agents are able to maintain higher rewards from the beginning of retraining since they keep a previous ability compared to retrained SAC agents (Figure 6). With SAC-I, we advance the use of SAC methods by introducing the use of multiple value networks with respective episodic replay buffers, as well as distinct entropy parameter estimation for skills at different training stages. In Figure 9, we show SAC-I is able to successfully learn the mixed version of the BipedalWalkerHardcore-v3, otherwise unsolvable with the standard SAC. We believe that the experiments and results presented in this paper offer a proof of concept of the advantages using SAC methods with inhibitory networks (SAC-I) for faster retraining.

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

# A   APPENDIX

## BROADER IMPACTS

Efficiency in retraining is useful in situations where training is computationally expensive or can cause damage and wear to instruments like robotic arms. Our work proposes a method of efficient retraining of agents by reusing trained policies and value function estimators. Ultimately, we contribute to the body of work that prioritizes creative reuse of models.

Table 1: List of common hyper-parameters used in the SAC training.

| Parameter | Value |
| --- | --- |
| Optimizer | Adam |
| Replay buffer size | 1.0e6 |
| Number of hidden layers | 1 (LunarLander), 2 (BipedalWalker) |
| Number of hidden neurons | 256 |
| Batch size | 64 and 128 |
| Learning rate | 5.0e-4 |
| Discount factor $\gamma$ | 0.99 |
| Soft $\tau$ | 1.0e-3 |
| Entropy target $\tau$ | -3 |
| Activation function | ReLU |
| Target update interval | 1 |
| Number of episodes before training $\pi_I$ | 0 (LunarLander), 100 (BipedalWalker) |

## SUPPLEMENTARY EXPERIMENTS AND RESULTS

**LunarLanderContinuous-v2 with bomb**. The observation space consists of the original 8 states (related to position, angle, velocities and leg contact) plus 4 values indicating the upper left and bottom right coordinates of the bomb zone (a box with a width of $\approx 10\%$ of the screen size). The agent assumes horizontal coordinates $x \in [-1, 1]$ and vertical coordinates $y \in [0, 1.4]$, where $y = 0$ corresponds to the ground. The bomb is randomly placed with center coordinates $x_c \in [-0.2, 0.2]$ and $y_c \in [0.1, 0.5]$, and appears with a default frequency of 50% (stop trials), which creates an environment with the highest uncertainty. In the stop trials, the bomb appears randomly after the agent achieves an altitude between the range $[0.9, 1.1]$. Importantly, the bomb coordinates are included in the observation state only after the bomb appears so that the agent does not know about its existence beforehand. If the bomb is absent, four dummy negative values are provided in the state. The episode ends if the agent hits the bomb (i.e. agent inside the bomb zone) or the ground, successfully lands or reaches the maximum length of 1000 steps.

**Conservative reward shaping**. In LunarLanderContinuous-v2 with bomb, it is expected that how the inhibitory reward $r_I$ is shaped, as described in the main text, influences the agents' training. Indeed, we tested a few shaping variants and observed differences. However, results comparing SAC-I vs. SAC were qualitatively similar to the ones presented so far. One potential issue with the current reward shaping to avoid the bomb is that it does not consider any of the aspects involved in landing. Therefore, we present another variant of the $r_I$ which includes some of the aspects of the raw reward such as rewarding getting close to the landing pad, slowing down, and penalizing "aggressive" maneuvers (large velocities and vertical angles). The more conservative reward shaping is given by the sum of the following components

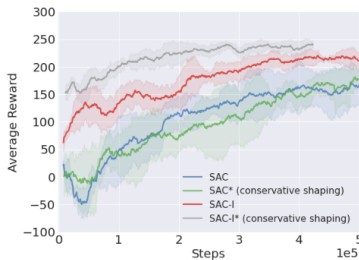

Figure 10: Comparison of SAC and SAC-I agents with conservative shaping.

$r_I = r_x + r_y + r_{angle} + r_{vel}$, where $r_x = -1/(6 \times d_x + 0.1) + 0.77$, $r_y = -3 \times (d_y - 0.05)^2$, $r_{angle} = -angle^2$, $r_{vel} = -2 \times (v_x^2 + v_y^2 + v_{angle}^2)$, and $(d_x, d_y)$ are the horizontal and vertical distances to the bomb, respectively. In short, the inhibitory reward has maximum value 0 and components with quadratic forms. Coefficients are adjusted to have $r_I < 0$ and a balance between the different rewards. If the inhibition rule given by conditions $d_x < 0.2$ and $y > y_b$ is met, the Q-I

network is used along with the inhibitory reward $r_I$. In this version, the bomb avoidance is driven by the horizontal distance $d_x$, independent of $d_y$, making the agent more conservative. Results are shown in Figure 10. Unlike SAC, the SAC-I is benefited by the shaping and it offers a mechanism to include additional shaping without interfering previously learned Q networks.

**BipedalWalker environments**. In both BipedalWalker-v3 and BipedalWalkerHardcore-v3 tasks, the primary goal is to reach the end of the track using the least amount of energy as possible. The agent gets a dense reward for moving forward (total 300+ points up to the end) and -100 for falling. There is a small cost for applying motor torque and moving the joints, so agents with optimal movements score more. The observation state has a total of 24 variables split between 14 related to the agent's body (including the angles and velocities of hull, hip joints, knee joints, vertical and horizontal speeds) and 10 LIDAR sensor readings. Everything the agent knows about the environment is through the sensors, so it does not know about obstacles or holes until they are scanned by the sensor, neither about its position in the environment since no coordinates are given. Note that sensor readings are relative to the agent's position and angle. These characteristics make the hardcore version significantly more challenging than the plain terrain version. The agent has 2 legs, each leg has 2 joints, and therefore the action space has size 4 with continuous values within the range of $[-1, 1]$. Each episode ends whenever the agent reaches the end of the track, falls or reaches the maximum of 2000 steps. Some informal discussion regarding the difficulty in solving the task can be found in https://ai.stackexchange.com/questions/13848/has-anyone-been-able-to-solve-openais-hardcore-bipedal-walker-with-their-implem .

**BipedalWalker experimental design**. The standard SAC agent is trained in the BipedalWalker-v3 to be used as a baseline agent in further training in the hardcore version. Agents are trained with three different seeds for 3K episodes, and the one with the best average reward is selected as the baseline. This agent reaches an average over the last 100 episodes above 300+ in about 0.5M steps similar to the results reported in Wei & Ying (2021). All the network weights are transferred to the retrained agent. The Q-I network is also retrained from previous weights. For the BipedalWalkerHardcore-v3, we train agents with five seeds for 5K episodes, and compute the average reward across runs.

**Effect of removing the fall penalty**. In neuroscience, punishments (negative rewards) are known to produce aversive conditioning or prediction of aversive events that leads to defensive or aggressive Pavlonian actions (Seymour et al., 2007), what in turn can cause fear of failure and procrastination in learning (Martin, 2012). Negative rewards are often successfully used in RL. However, there are studies showing that intermixing positive and sparse negative rewards might have adverse effects in learning, as reported in a recent work using DDPG in robotic environment Vargas et al. (2020). Moreover, optimally shaping reward is still an open area of research (Hu et al., 2020), and it is likely to depend on the particular DRL method being used.

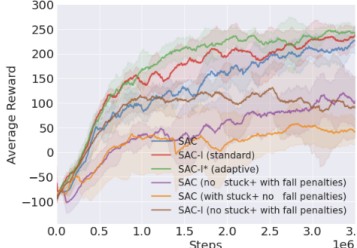

Interestingly, in our experiments in BipedalWalkerHardcore-v3, we observe that the SAC agents are greatly benefited from removing the fall penalty from the raw environment reward. We observe that, with the fall penalty, often the agent learns to simply stop in front of the block by fear of falling (which is likely a local minimum), never exploring enough to overcome the large obstacles. In Figure 11, we illustrate the effects of removing the fall penalty and including the stuck penalty. Removing the fall penalty (-100) included in original reward/penalty provided by the environment allowed the agent to explore riskier actions to overcome obstacles. The SAC agent without stuck penalty is also unable to learn the task successfully (purple line). It should be noted that retraining from a baseline agent is also critical to speed-up training and make possible solving the task. Average reward represents the average over the last 100 episodes of the raw environmental reward.

Figure 11: Agent performance in BipedalWalkerHardcore-v3, retrained using SAC algorithm with (brown) and without (blue line) the fall penalty.

**Training from scratch in the mixed version of the BipedalWalkerHardcore-v3**. In Figure 12, we show the average reward during training for the SAC and SAC-I agents, in the environment with 90% stop trials. Training the SAC-I agent from scratch is slower than retraining from the baseline agent (Figure 10, Stop trial 90%). However, the SAC-I agent is still able to solve the hardcore task (Stop trials). Meanwhile, the SAC agent is only able to solve the simple task (Go trials). The advantage

of retraining is evident during the initial training steps. The retrained SAC-I agent completes the simple task from the beginning of the training (compare with Figure 10, Stop trial 90%). The SAC-I agent trained from scratch learns to complete the Go trials only after 1M steps.

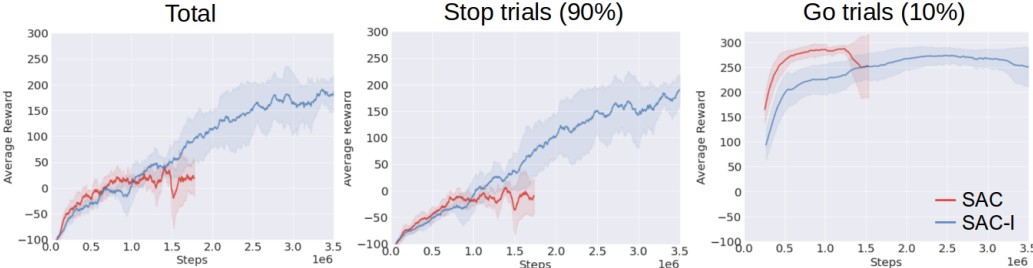

Figure 12: SAC and SAC-I agents performance in the mixed version of BipedalWalkerHardcore-v3. Agents are trained from scratch. Stop trials are 90% of the total number of episodes.

