# OpenReview forum: "Soft Actor-Critic with Inhibitory Networks for Faster Retraining"
_ICLR.cc/2022/Conference — ICLR 2022 Submitted_

### Official Review · Reviewer_y7rr · 2021-10-27

**Correctness:** 2
**Technical Novelty And Significance:** 2
**Empirical Novelty And Significance:** 2
**Recommendation:** 3
**Confidence:** 3

**Main Review:**

Overall, the proposed method is sound and the paper includes experiments that well illustrate the problem setting. However, I have several concerns regarding clarity of presentation, novelty of the contribution, and the experiments.

Regarding clarity, If I understand the paper correctly, the proposed method is essentially a hierarchical policy with discrete primitives that get activated one at a turn by a high level policy, called inhibitory network. As presented, the framework is limited to two primitives, called “Go” and “Stop” , though an extension to any number of primitives follows trivially. Many of the details still remain unclear to me (as I will explain below), and it is very possible that I have missed important ones.

In particular, I would suggest including a concrete example of the realization of the method, with the connection to hierarchical policies right in the introduction; and improving the Methods section by including more details and being more careful with the notation and claims. Currently there are many aspects that are not accurate or not addressed at all. Here are some examples:
* What is the exact role of the inhibitory network (e.g., what is $q$ in Figure 2)?
* Is the network learned, and if it is, how do you learn it and what is the objective for learning it?
* When the inhibitory network is learned, the split of the data between the two replay pools will be affected, or does it?
* In the first paragraph of page 9, it is explained that SAC-I* learns the inhibition rule by adjusting a reward weight parameter $w$. How does this relate to what is presented in Figure 2?
* What if $s$ and $s’$ belong to different partitions, in which replay buffer the corresponding transition goes?
* In Equation 1, conditional expectation depends on the variable $t$ that is visible only inside the expectation so it cannot be a conditioner.
* In the first paragraph of page 3, $\rho_pi$ is defined as the state marginal, though it should represent the dynamics.
* Are there separate policies for the Go and the Stop behaviors, or just a single one as in Figure 2? If there is only a single policy, then what is the purpose of having two temperatures $\alpha_R$ and $\alpha_I$ (Section 3.3)?

Regarding contributions, it is not clear how the approach is different from hierarchical RL with discrete low level behaviors. There is indeed a difference in how the data is split in two replay pools, each used to train a separate Q function, but a better justification is needed to explain why this difference is an important one. I think the work would substantially benefit from reformulating the approach as an HRL method, carefully pinpointing the improvements and contributions over existing HRL literature, and evaluating their relative importance. The connection to inhibition in the brain is interesting, but as presented at the moment, hides the connection to HRL.

Regarding the experiments, there is some mismatch between the figures and the text. For example, Figure 5 shows that training without the inhibitory network is more efficient. However, the text suggests the opposite. Similarly, Figure 9 suggests that pure SAC performs comparably to SAC-I* (adaptive), whereas SAC_I (standard) fails, but the text claims that “only the SAC-I agents are able to successfully learn the stop trials.”  Also comparing SAC to SAC-I is not completely fair as SAC-I makes use of domain knowledge in terms of handcrafted inhibition rule. Comparing to SAC-I* that learns the inhibition rule (if I got this right) is well justified, but no details are given regarding how it is learned.

Minor comments:
* The sentence: “ For states with high reward, a low entropy policy is permitted while, for states with low reward, high entropy policies are preferred, leading to greater exploration” is inaccurate. The magnitude of the reward and the entropy of a state are not related. Instead, the entropy of a state is determined by the relative magnitude of the expected future returns for different actions.
* Vague sentence, can you please reword: “Stop networks estimate the value of the unexpected event, leading the policy to learn new skills.”
* The experiment section mentions that “... the bomb coordinates are included in the observation state only after the bomb appears…”. How is this implemented in practice? Does the network allow a varying number of observations?


**Summary Of The Paper:**

The paper proposes inhibitory networks for reinforcement learning. Inhibitory networks choose a behavior among a set of low-level behaviors to be executed at the current state. The inhibitory network can take the form of a handcrafted rule (i.e., execute policy X if some state feature is active, else execute policy Y), or can be learned similarly to hierarchical policies. The method is implemented on top of soft actor-critic and evaluated in two OpenAI Gym environments, LunarLander and BipedalWalkerHardcore with some non-default modifications, and compared to a baseline standard SAC implementation.

**Summary Of The Review:**

The clarity of the paper is not sufficient for assessing its contributions and how it positions itself with respect to the prior literature.

---

> ### Author Response · Authors · 2021-11-19
> **Answers to Reviewer y7rr**
>
> We acknowledge the reviewer’s thorough review and comments. As suggested, reformulating and presenting the work from an HRL perspective would be helpful to make the contribution clearer and broader. However, we should mention that the goal of this paper was limited to highlighting the advantages of using dual value-functions to speedup the retraining of a single policy network. We did not aim to address the problem of solving a complex task by using multiple policy networks as usually done in HRL. Follow the questions and comments raised by the reviewer (“R”) and our answers (“A”). Hopefully, they clarify some of the concerns.
>
> **R10:** “What is the exact role of the inhibitory network (e.g., what is q in Figure 2)?”
>
> **A10:** The role of the inhibitory network is selecting or modulating the q-value (“q” in Figure 2) that goes to the policy network. As described in Section 3.4. “This network can be trained as an automated hard switch or a soft modulator between the regular and inhibitory Q networks.”
>
> **R11:** “Is the network learned, and if it is, how do you learn it and what is the objective for learning it?”
>
> **A11:** It can be learned as we describe in Sections 4.1 (Learning when to inhibit) and 4.2 (Learning how to inhibit). Its loss function should be the same as the policy network (Equation 3).
>
> **R12:** “When the inhibitory network is learned, the split of the data between the two replay pools will be affected, or does it?”
>
> **A12:** Correct, the inhibitory policy network (or alternatively an inhibition rule) will determine whether a sample will belong to the regular replay buffer ($D_R$) or the inhibitory buffer ($D_I$) as detailed in Algorithm 1.
>
> **R13:** “In the first paragraph of page 9, it is explained that SAC-I* learns the inhibition rule by adjusting a reward weight parameter w. How does this relate to what is presented in Figure 2?”
>
> **A13:** Indeed, the adaptive SAC-I is not represented in Figure 2. The weighting happens during the Q-I network updating (Equation 4). In BipedalWalkerHardcore, a simple inhibition rule ($r_{stuck} < 0$) was used to switch between $Q_R$ and $Q_I$.
>
> **R14:** “What if s and s′ belong to different partitions, in which replay buffer the corresponding transition goes?”
>
> **A14:** That’s a good question. The partitioning is driven by the current state s, not the later s’. So on the transition borders, the replay buffers might contain the next states s’ that do not “belong” to that partition.
>
>
> **R15:** “In Equation 1, conditional expectation depends on the variable t that is visible only inside the expectation so it cannot be a conditioner.”
>
> **A15:** Thanks for catching the inaccuracy. We have now corrected Equation 1.
>
> **R16:** “In the first paragraph of page 3, ρpi is defined as the state marginal, though it should represent the dynamics.”
>
> **A16:** Correct, ρpi represents the dynamics, i.e. the next state sampled from the transition probability. We updated the equation and the text to have a consistent notation.
>
> **R17:** “Are there separate policies for the Go and the Stop behaviors, or just a single one as in Figure 2? If there is only a single policy, then what is the purpose of having two temperatures αR and αI (Section 3.3)?”
>
> **A17:** There is a single policy network as shown in Figure 2. The purpose of having two temperatures is to weight the log-probability of the actions differently as described in Equation 6, ultimately allowing two different entropy vs. reward scaling, depending on the state.
>
> **R18:** “Regarding the experiments, there is some mismatch between the figures and the text. For example, Figure 5 shows that training without the inhibitory network is more efficient. However, the text suggests the opposite. Similarly, Figure 9 suggests that pure SAC performs comparably to SAC-I* (adaptive), whereas SAC_I (standard) fails, but the text claims that “only the SAC-I agents are able to successfully learn the stop trials.”.”
>
> **A18:** Thanks for catching the mistake. Indeed, those errors were inserted during the last pass of editing when we sought to keep a consistent color-coding across the different figures. For instance, red color should indicate SAC-I and blue color should indicate SAC methods.  This mistake made it such that in Figure 5, the colors were flipped in the plot. The one outperforming is SAC-I and should be red color. In Figure 9, colors in the legend were flipped. Blue is for SAC and red for SAC-I (standard) as in Figure 8. They are corrected in the updated manuscript.

---

> > ### Comment · Reviewer_y7rr · 2021-11-29
> > **Response**
> >
> > Thank you for clarifying most of my questions. The main issue that I still have is that paper seems to be missing the right context, including comparisons and discussions how it relates to prior work, and thus evaluating its importance and correctness is hard. Thus I feel that the paper is not yet ready for publishing.

---

### Official Review · Reviewer_dDqD · 2021-11-01

**Correctness:** 3
**Technical Novelty And Significance:** 2
**Empirical Novelty And Significance:** 3
**Recommendation:** 5
**Confidence:** 4

**Main Review:**

**Strong Points:**

This paper investigates a set of environments and tasks for multi-task / transfer learning. While these environments are fairly standard (from OpenAI gym), the tasks themselves, from what I can tell, are not widely used. I see these transfer tasks as a useful contribution of the paper, which may become adopted in future papers.

The paper appears to be technically correct, employing a modified version of SAC to investigate transfer learning tasks. The authors examine ablations of their proposed method to isolate the contributions from each aspect. The authors also vary aspects of the environment to investigate how this affects transfer performance.

The authors attempt to bring in motivation from inhibitory control in neuroscience. While I feel that this aspect needs improvement (see below), I appreciate the effort to port these concepts into deep RL and machine learning.


**Weak Points:**

The empirical evaluation is weak and could be improved in a number of ways. For instance, the main baseline used in the paper is pre-trained SAC. A more appropriate baseline would be to compare with alternative methods for multi-task or continual learning. Despite citing several lines of work in the introduction, e.g., various forms of hierarchical RL and compositional value functions, these other baseline are absent from the evaluation. Another weakness of the empirical evaluation is that SAC-I is evaluated on only two environments from similar domains. While I agree that these are interesting tasks, it may be helpful to also evaluate in settings used by previous works in continual RL, enabling a proper comparison. Finally, the evaluation entirely focuses on task performance. A more complete empirical evaluation would include further analysis on the mechanics underlying the inhibitory network and how it affects the resulting policy.

The method itself is difficult to follow, due the number of variations presented. Overall, the presentation of the method comes across as somewhat ad hoc. For instance, in some cases, an inhibition rule is defined, whereas in other cases, this is learned by a separate inhibition policy. In both environments, the authors use various redefinitions of the reward functions for shaping and learning the task. Separate replay buffers are used for the different tasks, along with separate temperature factors. The Q-value is fed into the policy network, which is an odd design choice that seems poorly motivated. In total, it’s difficult for the reader to assess where this method fits into the broader landscape of previous works and whether the proposed design choices are novel and/or reasonable. Fixing these issues would involve more of a first-principles approach to describing the issues with previous works in hierarchical / multi-task / continual RL and proposing and analyzing well-founded methods for overcoming these challenges.

There is a considerable amount of work in related areas of multi-task RL, non-stationary environments, online/continual learning, etc. that is largely overlooked. These are vast, interconnected areas, so I would not expect an extensive literature review. However, I would expect a more in-depth discussion beyond what is currently present in Section 2.4. This discussion would help to situate the proposed method.

There appear to be some inconsistencies in the empirical presentation. SAC outperforms SAC-I in Figure 5, although the accompanying text states otherwise. Similarly, SAC performs nearly as well as SAC-I* (adaptive) in Figure 9, again, in disagreement with the text. Note: the plotting colors in Figure 9 are also inconsistent with Figure 8. My guess is that these are both errors. These errors (and perhaps others) need to be fixed.

The authors motivate the proposed method by describing connections to inhibitory control, from neuroscience. While I’m generally supportive of neuroscience inspiration in machine learning, it’s not clear what these sections/paragraphs provide for the paper. That is, the details of SAC-I are so disconnected from the specifics of the neuroscience research that this motivation effectively amounts to ‘animals can switch between multiple tasks.’ This doesn’t strike me as any more helpful than the motivation from existing works in hierarchical RL, e.g., options. By framing this work in terms of neuroscience, it may needlessly complicate the presentation, raising a host of questions, such as whether separate replay buffers and the ability to accurately classify tasks are biologically valid. I would suggest that the authors either a) largely remove these sections or b) show more specifically how the inhibitory control literature can inform RL beyond what currently exists.

**Additional feedback:**

Section 2.1: This notation is imprecise / sloppy. For instance, in Eq. 1, the environment dynamics do no appear, and $r$ is not defined.

Section 2.2: Missing citation for PPO. It would be worth unpacking the last sentence (regarding KL minimization), or at least pointing to a source that describes this, such as Levine’s 2018 tutorial.

Section 2.3: It’s not clear that anterior cingulate cortex resembles TD learning. To my knowledge, this is more in-line with model-based estimation, not model-free.

Section 3.1: By conditioning the policy on $Q$, this effectively becomes a mixture of policies, which is equivalent to a hierarchical policy. Ideally, connections and differences from previous works in hierarchical RL should be discussed. It’s not clear that this diagram really matches Algorithm 1.

Section 4.1: It’s not clear what the reward shaping analysis really shows. It seems like the point is that SAC-I is less negatively affected by the shaping reward.

**Summary Of The Paper:**

This paper presents a method for transferring a pre-trained agent to a new task. The authors refer to this method as SAC-I, as they are combining ‘inhibition’ with soft actor-critic (SAC). The basic idea is to train a separate set of Q-networks and potentially an inhibitory policy network. These are then used with the existing policy. Various agents are presented on two Box2D environments from OpenAI gym. SAC-I generally improves over SAC.

**Summary Of The Review:**

Currently, the paper is lacking an adequate discussion of related works, as well as appropriate baselines beyond SAC. As the authors are proposing a new method for transfer learning, a comparison with existing works is necessary for proper empirical evaluation. Given that this method shares some similarities with hierarchical RL, these may be useful baselines. Looking to these baselines could also provide additional tasks and environments explored in previous works. Without these improvements, the proposed method is difficult to evaluate, as the larger context is missing. For these reasons, I cannot recommend acceptance at this time.

---

> ### Author Response · Authors · 2021-11-19
> **Answers to Reviewer dDqD**
>
> We acknowledge the reviewer’s thorough review and valuable comments. They will be helpful to improve the work. Writing suggestions were incorporated in the updated manuscript. Follow our answers (“A”) to the main concerns and questions raised by the reviewer (“R”).
>
> **R6:** “The empirical evaluation is weak and could be improved in a number of ways. For instance, the main baseline used in the paper is pre-trained SAC. A more appropriate baseline would be to compare with alternative methods for multi-task or continual learning. Despite citing several lines of work in the introduction, e.g., various forms of hierarchical RL and compositional value functions, these other baseline are absent from the evaluation.”
>
> **A6:** We agree that a more extensive evaluation would be beneficial and acknowledge the suggestions. In this work, we limited the scope to demonstrate the advantage of applying the inhibitory network's approach to SAC methods in cases where there is a need for retraining a previous agent. This work is not aimed to completely solve complex multi-task environments or address continual learning, neither it is combining previously learned skills similar to what is done in compositional value functions experiments. Therefore, we thought those would not be an equal comparison. An extension of our proposal would include comparisons with HRL and value-composition methods.
>
> **R7:** “The method itself is difficult to follow, due the number of variations presented... Fixing these issues would involve more of a first-principles approach to describing the issues with previous works in hierarchical / multi-task / continual RL and proposing and analyzing well-founded methods for overcoming these challenges.”
>
> **A7:** Thanks for the suggestion.
>
> **R8:** There appear to be some inconsistencies in the empirical presentation. SAC outperforms SAC-I in Figure 5, although the accompanying text states otherwise. Similarly, SAC performs nearly as well as SAC-I* (adaptive) in Figure 9, again, in disagreement with the text. Note: the plotting colors in Figure 9 are also inconsistent with Figure 8. My guess is that these are both errors. These errors (and perhaps others) need to be fixed.
>
> **A8:** Thanks for catching the error! Indeed, those errors were inserted during editing in which we sought to keep a consistent color-coding across the different figures. For instance, red color should indicate SAC-I and blue color should indicate SAC methods.  Thus, in Figure 5, the colors were flipped in the plot. The one outperforming is SAC-I and should be red color. In Figure 9, colors in the legend were flipped. Blue is for SAC and red for SAC-I (standard) as in Figure 8. They are corrected in the updated manuscript.
>
> **R9:** “The authors motivate the proposed method by describing connections to inhibitory control, from neuroscience. While I’m generally supportive of neuroscience inspiration in machine learning, it’s not clear what these sections/paragraphs provide for the paper... I would suggest that the authors either a) largely remove these sections or b) show more specifically how the inhibitory control literature can inform RL beyond what currently exists.”
>
> **A9:** We agree with the reviewer that, as it is presented, the association with inhibitory control from neuroscience is loose, and it deserves further investigation. In this paper, we sought to motivate the use of multiple (conflicting) value functions in light of how the brain processes conflict and implements inhibition.
>
> Thanks for the valuable suggestions. In fact, the presented approach using multiple value functions is general and could be applied to a more broad setting. Also, we believe that there are interesting mechanisms present in the brain that could be conceptually explored in RL research.

---

> > ### Comment · Reviewer_dDqD · 2021-11-27
> > **Response to authors**
> >
> > Thank you for the response and for fixing the plotting errors.
> >
> > Overall, my assessment of the paper has not changed substantially. The aim of this paper is in developing a system that can re-use previously learned policies while learning to perform modified versions of previous tasks. Naively applying current deep RL algorithms, e.g., SAC, does not generally perform well in these settings. Accordingly, I expect the authors to compare with previous works that explore a similar multi-task/continual/transfer aim. These baselines, as well as more thorough contextualization within the current literature, will help to improve the paper.

---

### Official Review · Reviewer_tm8g · 2021-11-02

**Correctness:** 3
**Technical Novelty And Significance:** 2
**Empirical Novelty And Significance:** 2
**Recommendation:** 3
**Confidence:** 3

**Main Review:**

**Strengths**

* Considering that RL is about sequential decision making and its importance in developing human-like agents, approaching the problem with a neuroscientific view might be a helpful and meaningful direction. I believe it could be a more considerable submission if the need for human knowledge is minimized and its empirical performance is improved (see my comments on the weaknesses below, please).

**Weaknesses**

* The concept of retraining should be more formal or specific.

  In Sec.2.4, the authors state that "we are primarily focused on transferring learned value and policy functions among identical aspects of a task, while learning new skills (value functions) and retraining the previous learned policy within the similar environment", but its difference from transfer learning in RL is still not clear, as usual transfer learning in RL also involves transferring learned components. I think "identical aspects of a task" doesn't clarify the dissimilarity.

  I suggest the authors to discuss which components of the original MDP can change and why and how retraining is different from common practices of transfer learning in RL.

* My biggest concern is that this work seems to require or rely on human engineering of reward functions too much (regardless of learning an inhibitory policy or not).

  Especially, it looks that regular rewards $r_R$ are designed to contain mixed, conflicting signals or no signals about dangers (e.g., bombs), while inhibitory rewards $r_I$ are configured to provide meaningful signals about the dangers. For instance, in the paragraph "Advantage of retraining." of Sec.4.1 for the LunarLanderContinuous experiments, the authors state that the penalty, $r_{bomb.proxy}$, which is defined using the agent's distance to the bomb, is combined with the original reward and given to the agent for the retraining. Given that the bombs appear close to the landing pad (Fig.3), the regular rewards $r_R$ would be a mixture of two competing signals. On the other hand, the inhibitory rewards $r_I$ defined in the paragraph "Effect of using episodic memory and dual alpha in SAC-I." are just $r_{bomb.proxy}$, which means the SAC-I agents receive the clean, decoupled signals about the bombs.

  I'm aware that the authors suggest it as part of the problem setup, but assuming the specific problem, I think the contribution of this work is not very significant. Compared to the normal case with a single reward function, this problem setup requires defining and adjusting two reward functions properly, which is not free (i.e., humans' job) and can be more complex.

  I believe that removing or minimizing the requirement of manual reward function engineering could be a good way to improve this work.

* Little more details about the inhibitory policy would be nice. I would like to see clarification especially on the following points: Does it output binary actions to choose between regular and inhibitory modules? What exactly does "soft modulator" from Sec.3.4 mean?

* The performance improvement over SAC seems to be marginal in many cases, considering the shaded area of the plots. Increasing the number of runs might help make the results more clear, but I also suggest finding ways to improve its performance.

**Minor writing suggestions**

* Other methods use multiple "policy" -> "policies" (Sec.1)
* First, we develop the SAC-I architecture for accelerated retraining, "that" encompasses -> "which" (Sec.1)
* which is not able to "successfully solve the task" -> "solve the task successfully" (Sec.1 and similar occurrences)
* "previous" learned -> "previously" (Sec.1, Sec.2.4)
* Broadly speaking, transfer learning in RL consists of "transfering" the knowledge -> "transferring" (Sec.2.4)
* to improve the learning performance in a related, but "different, task" -> "different task" (Sec.2.4)
* "over estimation" -> "overestimation" (Sec.3.1)
* The inhibitory policy network is a "stand alone" agent -> "stand-alone" (Sec.3.4)
* with "particular focus" -> "a particular focus" (Sec.5)

**Summary Of The Paper:**

The authors present SAC-I, which is a modified version of SAC designed for retraining of some existing agent on an updated environment or task. Inspired by the concept of inhibitory control from neuroscience, they learn a separate Q functions and optionally a policy for inhibition. They define some inhibitory reward $r_I$ for each environment to enable it, and make comparison with SAC on modified versions of LunarLanderContinuous and BipedalWalkerHardcore-v3.

**Summary Of The Review:**

Although I appreciate the attempt to bring another neuroscientific concept into RL, I'm mainly worried about the significance of the reward function engineering in the presented results and thus a smaller importance of the proposed method.

---

> ### Author Response · Authors · 2021-11-19
> **Answers to Reviewer tm8g**
>
> We are very thankful for the reviewer's time and comments. We acknowledge many of the suggestions by the reviewer, but we believe that the main criticism regarding human reward engineering is unfair. The final SAC-I implementation does not use/require additional reward shaping and uses the exact same reward function as the SAC agent. We apologize for not being clear in the text and updated the document to be more explicit. We have also incorporated the writing suggestions in the updated manuscript. Details are given as follows, where “R” indicates the reviewer’s comment and “A” indicates our answer.
>
> **R5**: “My biggest concern is that this work seems to require or rely on human engineering of reward functions too much (regardless of learning an inhibitory policy or not)…”
>
> **A5**: We understand the concern but disagree with the statement. Indeed, as mentioned by the reviewer, reward engineering was used in the initial results presented in Figure 4 to show a use case of SAC-I. However, this is not the main implementation, nor a requirement. Later, in Figure 6 and Section 4.2 (page 8), we show the results for the SAC-I implementation without any reward shaping (for both cases with a given inhibition rule or using a fully automated switch, i.e. inhibitory network). Both SAC and SAC-I agents use the same reward function $r = r_0 + r_{bomb}$, where $r_0$ represents the original environment reward and $r_{bomb}$ is a sparse penalty (which happens only when the agent hits the bomb). The only difference is that the SAC-I agent will keep a Q network with the original reward $r_0$ and another Q-I network to learn the new reward $r = r_0 + r_{bomb}$. Similarly, for the BipedalWalkerHardcore task, as described in Section 4.2, both SAC and SAC-I agents use the same reward function given by $r = r_0 + r_{stuck} – r_{fall}$. Thus, none of the SAC-I agents presented in Figures 8 and 9 use any type of advantageous reward shaping as compared to the SAC agents as stated by the reviewer.
>
> To reiterate, in LunarLander with Bomb, reward engineering is used as an example but it is not necessary for the SAC-I to work. The only modification in the environment was to include a bomb penalty. In the BipedalWalkerHardcore example, there is no additional reward shaping other than the ones necessary to solve the task with a simpler non-recurrent architecture, i.e., adding the stuck penalty $r_{stuck}$ and removing the fall penalty $r_{fall}$ in both SAC and SAC-I agents. The statement “I'm mainly worried about the significance of the reward function engineering in the presented results and thus a smaller importance of the proposed method.” along with others are unfair because it refers to secondary results presented in Figure 4, while it does not apply to the main results presented in Figures 6, 8 and 9, in which there is no additional reward shaping favoring SAC-I agents.

---

### Official Review · Reviewer_6ibM · 2021-11-03

**Correctness:** 3
**Technical Novelty And Significance:** 3
**Empirical Novelty And Significance:** 3
**Recommendation:** 3
**Confidence:** 4

**Main Review:**

The idea of two independent policies is interesting, but the use of the words ‘inhibitory’ and ‘stop-policy’ is a bit confusing. I’m not sufficiently familiar with the neuroscience literature to judge this well, but my understanding was that inhibition is about the suppression of reflexive actions, rather than about what to do instead. The agent presented here just has to choose between two policies.

One question that the two-policy situation raises is why to stop at two. Depending on the task, it might be that a different number is optimal. Another question relates to the effectiveness of the extra learning capacity (weights & updates): what if that capacity of the decider network and the second policy was allocated to a single policy instead? I could imagine that the factorization employed here can be more effective, although that probably depends on the environment, but it would be good to know instead of imagining. The experiments are done in environments that effectively seem to have two tasks, which might be why having two policies works well in practice. But what if the environment mixes three tasks? Or 20?

The method presented here is strongly reminiscent of the options framework - which is being cited - with a different approach to choosing start and stop states. It would help place the paper more clearly in the overall RL literature if this similarity and difference was discussed more explicitly.

It is not entirely clear to me what is happening in the case called adaptive SAC-I; is there an extra network being trained to estimate the weight $w$? And if so, what is the loss for this network?

Could the authors comment on the use of SAC as a starting point / base training method? I understand that it is a very effective maximum entropy method, but it is not necessarily the state of the art in RL tasks in general. Both the motivation from inhibition in the neuroscience literature, and the effective implementation of a two-policy agent are fairly applicable to many reinforcement learning techniques, not just SAC.

Other questions and remarks:
- Figure 4: how are the agents trained if they don’t have a replay buffer (episodic memory)? Algorithm 1 seems to indicate that no training will happen without.
- Apart from ‘inhibition’, the term ‘episodic memory’ is also a bit of a misnomer when seen against the use of that term in the literature: here it is just used as a replay buffer, if I understand the paper correctly. There is no notion of retrieving information from the memory within a given step here, which is commonly the case when the term ‘episodic memory’ is used.

On the whole, the paper does not entirely convince me that the proposed architecture will generalize well. Including experiments on a wider range of environments would address that concern. Including other baselines than SAC would also help to clarify the value of the contribution.


**Summary Of The Paper:**

This paper introduces an agent architecture with two independent policy networks (called ‘go’ and ‘stop’) and an assignment mechanism consisting of a network or a rule that activates one of the two every step. The two policy networks are in principle both soft actor-critic (SAC) agents, although the authors explore several variations on the degree of independence between them, with e.g. hyperparameters being tied in some cases. Each policy network is trained on replayed data from the subset of states that has been assigned to it by the assignment mechanism.


**Summary Of The Review:**

The proposed architecture is potentially interesting, even if it seems to go against Sutton’s “Bitter Lesson” by specializing an architecture for a specific case, rather than relying on learning. It does look brittle due to the hardwired nature of the choices that are being made; in particular the number of policy modules in the agent and the division of states across the policies. I am open to being convinced these aspects don’t make the agent brittle, and would raise my score in that case, but that would require more extensive experimentation and baselines.

There are some misnomers, or uncommon uses of terms: mainly ‘inhibition’, but also ‘episodic memory’. Otherwise the paper is well-presented; changing the terminology to be more in line with the literature would make it mostly easy to read.

My general recommendation at this point is to review the agent architecture and see if this is a reasonable point to present results. It might be interesting to see what this kind of structure does to agent performance on a wider array of tasks, and how those tasks are solved. If it does well generally, this would be a strong paper, but in its current form it looks a bit unfinished.

---

> ### Author Response · Authors · 2021-11-19
> **Answers to Reviewer 6ibM**
>
> We are very thankful for the reviewer's time and comments. Although we acknowledge the raised concerns, we believe that unfortunately there is a fundamental misunderstanding about the work, and would like to clarify. In short, this paper is not about training two independent policy networks (“go” and “stop”) as stated by the reviewer. As mentioned in the Introduction, “In this paper, we propose to address the problem through the use of multiple value functions to provide a complex evaluative input to a single policy network. By applying different value functions in a state-dependent fashion, the reward provided to the policy network during training can remain the same as in prior training when appropriate, and can switch to a different reward when the situation indicates new constraints or goals.”. Unlike previous option learning frameworks, our goals is not training multiple policies. Instead, we propose training an additional value network (“stop”) for the new reward/constrain component within a retraining context. Thus, the previous “go” value network is combined with the new “stop” value network, and used to update the single policy network, ultimately to achieve faster retraining. As shown in Figure 2, we introduce the use of an “inhibitory” policy network to select the value to be passed to the policy network, not to replace it.
> Here, we aim to answer other concerns. “R” indicates the reviewer’s comment, and “A”, the answer.
>
> **R1:** “The idea of two independent policies is interesting, but the use of the words ‘inhibitory’ and ‘stop-policy’ is a bit confusing. I’m not sufficiently familiar with the neuroscience literature to judge this well, but my understanding was that inhibition is about the suppression of reflexive actions, rather than about what to do instead.”
>
> **A1**: As mentioned in the initial comments, we do not propose the use of two independent policies. Instead, we proposed the use of two independent value functions. In neuroscience literature, “inhibition” is used at many levels going from synaptic connections to motor control and executive functions. We use it within the context of the latter, which is an established concept in cognitive neuroscience. As described in Section 2.3, “Inhibitory control, also known as response inhibition, is a critical component of the executive functions and refers to the ability to modify ongoing actions in response to unexpected and dynamically changing task demands (Aron, 2007; Diamond, 2013).”. The response inhibition is present not only in motor tasks, but in many cognitive tasks and associated with various clinical conditions such as addiction and ADHD (Goldstein and Volkow, 2002, Wodka et al., 2007).
>
> Additional references:
>
> (Goldstein and Volkow, 2002) Goldstein, Rita Z. and Volkow, Nora D. Drug Addiction and Its Underlying Neurobiological Basis: Neuroimaging Evidence for the Involvement of the Frontal Cortex. American Journal of Psychiatry 2002 159:10, 1642-1652.
>
> (Wodka et al., 2007) Wodka EL, Mahone EM, Blankner JG, Larson JC, Fotedar S, Denckla MB, Mostofsky SH. Evidence that response inhibition is a primary deficit in ADHD. J Clin Exp Neuropsychol. 2007 May;29(4):345-56.
>
> **R2:** “One question that the two-policy situation raises is why to stop at two. Depending on the task, it might be that a different number is optimal…”
>
> **A2:** Correct, there is not a particular reason to stop at two. The idea of using multiple value functions could be extended as needed by the task. In this work, we aimed to showcase the use of an additional “stop” value network to withhold the learned evaluation by the “go” value network. Our primary goal is not proposing a general algorithm to simultaneously learn multiple skills necessary to solve a complex task. Instead, learning a complex task by steps, in a modular fashion, as common in real-world problems. In many RL applications, there is a need/wish to retrain/reuse previously learned agents to add an additional component (for example, avoiding a new obstacle). We present a more efficient way to retrain the policy network by training an additional value network (“stop”), which we called “SAC-I”, instead of simply retraining the previous value network (“go”) to learn the stop-value, which we called “SAC” in the figures. Although we evaluate the method with two value functions, the proposed approach is general enough to be extended to multiple value networks, conditional to the number of changes in the environment. Indeed, this work was motivated by a real-world application and we believe that the approach might be helpful to many tasks requiring retraining previously learned policy network weights.

---

> > ### Author Response · Authors · 2021-11-19
> > **Answers to Reviewer 6ibM (cont.)**
> >
> > **R3:** “It is not entirely clear to me what is happening in the case called adaptive SAC-I; is there an extra network being trained to estimate the weight w? And if so, what is the loss for this network?”
> >
> > **A3:** Correct, as described in Section 3.4, an inhibitory policy network is “trained as an automated hard switch or a soft modulator between the regular and inhibitory Q networks… The loss functions are the standard ones as defined in Equations 2 and 3. ”. In the BipedalWalker experiments, the inhibitory policy network is trained to dynamically estimate the weight w for the stuck penalty (soft modulation), leading to the expression $r_I = r_0 + w * r_{stuck} – r_{fall}$. We called this adaptive SAC-I.
> >
> > **R4:** “Could the authors comment on the use of SAC as a starting point/base training method? I understand that it is a very effective maximum entropy method, but it is not necessarily the state of the art in RL tasks in general.”
> >
> > **A4:** We chose SAC because it is a very effective and simple maximum entropy framework, and offers a way to automatically estimate the temperature parameter. However, as mentioned by the reviewer, the approach is applicable to any other actor-critic method using a replay buffer. This work contributes by proposing the use of a dual value network approach (within the retraining context) to update a single policy network, and by demonstrating its effectiveness in the SAC framework.

---

### Decision · Program_Chairs · 2022-01-20

**Decision:**

Reject

**Comment:**

The paper proposes a hierarchical policy architecture with two substituent policies, "go" and "stop", and a controller mechanism for switching between them on every step (either a rule or a learned network), taking inspiration from neuroscience concepts of inhibition. Both are trained via Soft Actor Critic on the subset of states assigned to them and comparisons are made against a baseline. The use case targeted is the repurposing of pre-trained agents to new or updated environments.

Reviewers regarded the method as sound, technically correct, and involving illustrative experiments (although perhaps picking problems too carefully adapted to the solution being presented), and were positive on the general direction of taking inspiration from neuroscience. Reviewer y7rr found the details unclear, recommended more focus on a concrete realization of the general method, and questioned the differences with more traditional hierarchical RL; while many specific inquiries were addressed the reviewer's broad concerns about contextualization remained. Reviewer dDqD had similar concerns around confusing presentation and positioning within the broader literature on "multi-task RL, non-stationary environments, online/continual learning, etc.", and the discussion unfolded similarly -- many specific concerns addressed but fundamental issues remaining. 6ibM, like y7rr, raised the question of why one should stop at 2 policies rather than N policies, noted the under-discussed relationship to options, and questioned the starting point of SAC, and while this was clarified to be about value functions rather than policies, the reviewer still thought this was an ill-justified choice that rendered the system "brittle", and remained unhappy with baseline choices not extending beyond SAC-based agents. tm8g had similar concerns about clarity and in particular that reward engineering seemed central; the authors clarified that this was not the case.

There is wide consensus among qualified reviewers that the presentation (and in particular situating the method with respect to prior work) is inadequate for publication, and I am inclined to agree. As y7rr put it, "evaluating its importance and correctness is hard" without adequate context on the relationships to in particular existing work on hierarchical, multi-task and continual learning. While the direction appears interesting, unfortunately the hard work of contextualizing one's contribution is an utterly essential part of the scientific enterprise; without it we risk retreading well-explored terrain while merely wearing slightly different boots. I encourage the authors to further clarify their presentation incorporating the valuable feedback from the reviewers on this aspect.